# The Epigenetic Modifiers HDAC2 and HDAC7 Inversely Associate with Cancer Stemness and Immunity in Solid Tumors

**DOI:** 10.3390/ijms25147841

**Published:** 2024-07-17

**Authors:** Kacper Maciejewski, Marek Giers, Urszula Oleksiewicz, Patrycja Czerwinska

**Affiliations:** 1Undergraduate Research Group “Biobase”, Poznan University of Medical Sciences, 61-701 Poznan, Poland; maciejewskikacper@mensa.org.pl (K.M.); marek.giers00@gmail.com (M.G.); 2Department of Cancer Immunology, Poznan University of Medical Sciences, 61-866 Poznan, Poland; 3Department of Diagnostics and Cancer Immunology, Greater Poland Cancer Centre, 61-866 Poznan, Poland

**Keywords:** HDAC, cancer stemness, immunotherapy, tumor immunity, transcriptomics, TCGA

## Abstract

Dysregulation of histone deacetylases (HDACs) is closely associated with cancer development and progression. Here, we comprehensively analyzed the association between all HDAC family members and several clinicopathological and molecular traits of solid tumors across 22 distinct tumor types, focusing primarily on cancer stemness and immunity. To this end, we used publicly available TCGA data and several bioinformatic tools (i.e., GEPIA2, TISIDB, GSCA, Enrichr, GSEA). Our analyses revealed that class I and class II HDAC proteins are associated with distinct cancer phenotypes. The transcriptomic profiling indicated that class I HDAC members, including HDAC2, are positively associated with cancer stemness, while class IIA HDAC proteins, represented by HDAC7, show a negative correlation to cancer stem cell-like phenotypes in solid tumors. In contrast to tumors with high amounts of HDAC7 proteins, the transcriptome signatures of HDAC2-overexpressing cancers are significantly enriched with biological terms previously determined as stemness-associated genes. Moreover, high HDAC2-expressing tumors are depleted with immune-related processes, and HDAC2 expression correlates with tumor immunosuppressive microenvironments. On the contrary, HDAC7 upregulation is significantly associated with enhanced immune responses, followed by enriched infiltration of CD4+ and CD8+ T cells. This is the first comprehensive report demonstrating robust and versatile associations between specific HDAC family members, cancer dedifferentiation, and anti-tumor immune statuses in solid tumors.

## 1. Introduction

Solid tumors are biologically complex structures with substantial intratumor heterogeneity that contain transformed cancer cells, supportive cells, and tumor-infiltrating cells [1]. Also, tumor cells significantly differ in the potency to initiate and maintain tumor growth, with stem-like cancer cells exhibiting the highest self-renewal potency, responding rapidly and flexibly to environmental challenges and becoming a primary source of drug-resistant tumor cells [2]. Stem cell-associated molecular features of cancer cells might be acquired by the bulk tumor cells, i.e., in response to exogenous stimuli. Later on, those cells may experience phases of transitions between stem-like and non-stem-like states without any additional genetic perturbations. This phenomenon of stemness acquisition and maintenance robustly contributes to cancer cell heterogeneity, ultimately warranting drug resistance and tumor relapse [3].

Previous studies aimed to characterize the populations of cancer cells with stem-like properties (known as cancer stem cells, CSCs) in diverse tumor types, including breast [4], lung [5], liver [6], pancreatic [7], gliomas [8], melanomas [9], and many other tumors [10]. The stem cell-associated molecular features of solid tumors, also known as cancer stemness, are essential for cancer progression [11]. Hopefully, targeting factors that mediate cancer stemness might help overcome therapy resistance and improve clinical outcomes for cancer patients [12]. The negative association of cancer stemness with antitumor immunity is an essential concern in designing therapeutic approaches. As reported by Miranda A. et al. [13], solid tumors that exhibit high stem cell-like molecular traits are significantly abrogated with immune cells, and cancer stemness confers immunosuppressive properties on tumors. Accordingly, immunologically cold microenvironments can arise due to the presence of stem cell-like cancer cells, suggesting that by inducing cancer cell differentiation (irreversibly disrupting cancer stemness), tumors might become more susceptible to immunotherapy [13].

Epigenetic perturbations have a grounded role in mediating cancer development and progression, and at least partially, the dysregulation of epigenetic mechanisms facilitates the self-renewal of cancer cells [14]. Our recent reports highlighted the involvement of several epigenetic factors in the acquisition of cancer dedifferentiation [15,16]. As epigenetic mechanisms are highly reversible, they appear significantly responsible for cancer stemness acquisition or maintenance [17]. Among other epigenetic factors, the components of histone-modifying machinery are potent therapeutic targets due to their enzymatic activity that could be blocked with specific small molecule inhibitors [18].

Histone deacetylases (HDACs), a class of enzymes that remove acetyl groups from ε-N-acetylated lysine residues on target proteins—being either histones or non-histone targets—act primarily as epigenetic regulators of gene expression by modulating chromatin compaction [19]. Contrary to histone acetyltransferases (HATs), HDACs’ activity tightens the associations of histone tails with DNA and prevents the binding of transcription cofactors to DNA, resulting in gene repression.

There are 18 mammalian HDACs grouped into four distinct classes based on their sequence similarity, chemical structure, and cofactor dependency (Table 1) [20,21]. The class I HDAC group, which comprises HDAC1/2/3 8, is primarily localized in the nucleus and is ubiquitously expressed. On the other hand, class II (HDAC4/5/6/7/9/10) and class IV HDACs (HDAC11) can shuttle between the nucleus and cytoplasm and exhibit greater tissue-specific expression. All members of class I, II, and IV HDACs are zinc-dependent enzymes in contrast to class III HDACs (sirtuins), which constitute a structurally distinct subfamily that requires NAD+ to catalyze its activity [20,21]. Class I HDACs show the most robust histone deacetylase activity, while the remaining classes prefer other substrates. The specificity of HDACs for regulating distinct expression profiles depends on the cell type and the different partner proteins typically expressed in that cell, in addition to the signaling context of the cell. Specifically, HDAC1 and HDAC2 are catalytic subunits of the Sin3, Mi-2/NurD, and CoREST complexes, whereas HDAC3 is mainly recruited by the N-CoR/SMRT complex (characterized in detail in [20]). HDAC8 has not been described as a member of any protein complex so far. As for class II HDAC proteins, the recruitment into the multi-protein corepressor complexes does not promote their deacetylase activity but, instead, seems to have a protein scaffold role [20,21]. HDAC11, the only member of class IV HDACs, is the most recently discovered HDAC enzyme with a very short half-life (approx. 4 h), and a very efficient long-chain fatty acid deacylase activity [22].

Existing studies on the role of HDACs in facilitating cancer stem cell-like properties primarily focus on designating the involvement of one/several HDAC member(s) in one tested tumor type and do not allow for a broader perspective and a direct comparison of HDACs’ relation to tumor dedifferentiation. Here, we investigated the relationship between the expression of all HDACs and cancer stemness across various solid tumor types. We utilized data primarily from TCGA along with several free and open-access bioinformatic tools such as GEPIA2, TISIDB, GSCA, Enrichr, and GSEA, leveraging the idea of deriving new meaningful findings only from already existing tools and publicly available datasets.

Our findings represent the first comprehensive analysis, revealing robust and versatile associations between specific members of the HDAC family, cancer dedifferentiation, and the anti-tumor immune status within solid tumors. Our data strongly suggest that by targeting specific HDAC members, we may be able to render stem cell-like cancer cells more susceptible to immunotherapy. 

## 2. Results

### 2.1. The Expression of HDAC Family Members in Tumor and Normal Adjacent Tissues and the Association with Cancer Patients’ Survival

Firstly, we analyzed the differential expression of all histone deacetylase (HDAC) gene family members in malignant tissues of selected solid tumors (Table 2), relative to adjacent normal tissues using the TCGA [24] and GTEx data [25], respectively (Figure 1). We observed a significant upregulation of class I HDACs (HDAC1/2/3/8) in malignant tissues regardless of the tumor type. On the other hand, class IIA and IIB HDACs (HDAC4/5/7/9 and HDAC6/10, respectively) exhibit both up- and downregulation in tumors depending on the tumor type. Similarly, class III HDACs (sirtuins) and the only member of class IV (HDAC11) are differentially expressed in malignant tissues with prevalent upregulation across TCGA tumor types.

As shown in Appendix A, mutations of HDAC family members occur occasionally in selected studies when combined. They ranged from 1% up to 3% of total profiled samples for each gene, with the highest fraction of altered samples present in KIRC for HDAC3 (amplification in 13.94% of cases) and in PRAD for HDAC2 (deep deletion in 13.03% of cases). The expression of HDAC genes differs across solid tumors (Appendix A) and does not exhibit particular co-expression patterns (Appendix A).

Also, the expression of HDAC family members is scarcely correlated with patients’ outcomes in TCGA data, and few significant associations are strictly tumor-dependent and do not exhibit any consistent pattern within HDAC classes (Figure 2). The expression of HDAC members in tumors like KIRC and PAAD is correlated with only better prognosis wherever the trend is statistically significant for either overall survival (OS) or disease-free survival (DFS). On the other hand, tumors like LIHC, LUSC, PRAD, CEST, and STAD present opposite trends, being correlated with only worse prognoses. However, when using the Prognoscan platform [26]—an extensive collection of publicly available cancer microarray datasets with clinical annotation—we observed that when statistically significant, the higher expression of class I HDAC members is predominantly associated with worse survival. In contrast, the upregulation of class IIA HDAC proteins correlates with better prognosis (Appendix A). As for class IIA, class III, and class IV HDACs’ expression, the associations with overall survival are gene-specific and do not follow any particular trend.

### 2.2. The Expression of HDAC Family Members Is Associated with Clinicopathological Features of TCGA Solid Tumors in a Cancer-Dependent and Gene-Specific Manner

Next, we studied correlations between the expression of each HDAC family member and clinical features like staging and grading for every solid tumor, wherever such data were available in the TCGA database. As presented in Figure 3, HDAC family members’ expressions correlate with tumor stage or tumor grade in a cancer-dependent and gene-specific manner and do not follow any specific trends, even within each HDAC family class. It is worth noticing that the expression of HDAC1, HDAC2, and HDAC3 (all members of class I) significantly correlates with TGCT’s staging most robustly among all correlations in a positive manner (Figure 3B).

We further explored the associations between HDAC family gene expression and clinicopathological features by comparing patients with low and high expressions of specific genes (with 25th and 75th percentiles as cut-offs, respectively). We found statistically significant associations between the expression of multiple HDAC family members and TNM classifiers (i.e., tumor size, lymph node, and metastasis status) in several solid tumors (Appendix A). For example, the expression of HDAC2/3/7/10 and SIRT2/3/6/7 associates with tumor size in BRCA, the level of HDAC4/5/8/10/11 and SIRT1/5/7 associates with tumor size in KIRC, and the upregulation of HDAC2/4/6/7 and SIRT1/2/6 associates with tumor size in LUAD.

While tumor size differs between high- and low-expression groups for many HDAC members in many tumors, KIRC is the only one in which the expression of nine HDAC family members, HDAC1/5/8/10/11 and SIRT1/5/6/7, is associated with metastasis. Interestingly, a low expression of HDAC11 was associated with advanced disease status in KIRP, which complies with a statistically significant negative correlation between HDAC11 expression and both staging and grading in this tumor.

### 2.3. Class I HDAC Family Members Correlate Positively, While Class IIA HDAC Genes Correlate Negatively with Cancer Stemness across TCGA Solid Tumors

As previously reported, solid tumors display distinct levels of cancer stemness [11]. Here, we analyzed the association between the expression of HDAC family members and the level of tumor stemness, quantified with the transcriptome-based stemness index-mRNA-SI, as previously defined by Malta T. et al. [11]. We further validated our results with additional stem cell-derived gene signatures [27,28,29,30]. As presented in Figure 4 and Figure 5A, class I HDAC family members correlate positively, while class IIA HDAC genes correlate negatively with cancer stemness across TCGA solid tumors, with the most robust and most consistent associations observed for BLCA, BRCA, COAD (or effectively COADREAD), HNSC, LIHC, LUAD, LUSC, OV, PRAD, and STAD (Figure 4 and Appendix A). Specifically, HDAC2 (member of class I) and HDAC7 (member of class IIA) genes show the highest correlations across studied solid tumors, exhibiting statistically significant opposite trends for all the applied stemness indices. We observed only minor exceptions for HDAC7 in HNSC, LUAD, LUSC, and OV, where correlation is statistically significant for at least one of the applied stemness indices (Figure 4 and Appendix A). For HDAC2 and HDAC7 genes, the opposite statistically significant correlation trend is not observable in the case of LIHC, PRAD, and SARC. 

Previously, Malta et al. [11] have found a strong association between the mRNA-SI and known clinical and molecular features of TCGA BRCA tumors, demonstrating that the basal subtype, known to exhibit an aggressive phenotype associated with an undifferentiated state, displays the highest levels of mRNA-SI. Therefore, we analyzed the expression of class I and class IIA HDAC members in individual TCGA BRCA samples stratified by molecular subtype (PAM50). We observed a significant upregulation of HDAC2 in highly dedifferentiated basal and HER2+ subtypes, in contrast to other class I HDAC members, which exhibit equable expression patterns across BRCA subtypes. On the other hand, HDAC7 was significantly overexpressed in less aggressive luminal A and luminal B BRCA subtypes (Figure 5B,C), distinguishing from the expression profiles of other class IIA HDAC members. When testing the level of individual pluripotency markers’ expressions (namely OCT4 (POU5F1), SOX2, NANOG, and MYC) in highly stem cell-like TGCT tumors, we observed significant positive associations with class I HDAC members, with the most robust ones for HDAC2 (Figure 5D and Appendix A). This further supports our first observation that HDAC2 is strongly associated with cancer stemness. 

As HDAC2 and HDAC7 genes are observed to present the purest correlation trends with stemness indices in previously selected tumors, they were used as representative members of class I and class IIA for the subsequent analyses. 

We used the GSEA [31] to compare the HDAC2 and HDAC7-associated transcriptome profiles with a priori-defined stemness-associated gene signatures like Muller Plurinet, Wong ESC, Kim Myc, and Assou ESC, as previously described [27,29,30,32]. We confirmed significant enrichment of HDAC2-associated transcriptome profiles (Figure 6A–E) followed by significant depletion of HDAC7-related transcriptome profiles (Figure 6F–J) with stemness markers in all tested tumors. This strongly supports our first observation that HDAC2 is positively and HDAC7 is negatively correlated with tumor stemness.

Further, using the Enrichr tool [33], we performed the enrichment analysis to detect potential targets for known transcription factors in the HDAC2-related (Appendix A) or HDAC7-related (Appendix A) gene expression profiles. We identified BMI1, MYC, NANOG, and OCT3/4 (POU5F1 gene) pluripotency markers as the most prominent transcription factors associated with HDAC2 gene expression, highly supporting HDAC2’s association with cancer stemness.

### 2.4. HDAC2-Associated Transcriptome Profiles Are Significantly Enriched with Stemness-Related Hallmarks of Cancer, While HDAC7-Associated Transcriptome Profiles Are Enriched with Immune-Related Terms

Cancer stemness is significantly negatively associated with tumor immunity, with high stem cell-like tumors exhibiting low infiltration levels and significant immunosuppressive microenvironments [13]. It was unsurprising that HDAC2-associated transcriptome profiles were significantly depleted with immune-related signaling pathways or terms as determined with the GSEA using Hallmark’s collection of gene signatures (from MSigDB Hallmark 2020). As shown in Figure 7, the transcription profile associated with HDAC2 is highly enriched, while associated with HDAC7 is highly depleted with stemness-related terms like MYC targets, G2/M checkpoint, and E2F targets. These terms are statistically significant in all 11 selected solid tumors. On the other hand, the HDAC7 transcription profile is highly enriched with immune-related Hallmark terms, especially in SARC and HNSC.

Enrichment of systematically selected gene sets from GO Biological Process ontology collection (C5:BP, see Section 4.5) further confirmed our findings. As shown in Appendix A, ontology processes having immune-related ancestors are significantly enriched in HDAC7-related transcriptome profiles in SARC and HNSC, and to a lesser extent in STAD. Albeit with no statistical significance, these processes also exhibit enrichment in other tumors: PRAD, LUSC, LIHC, OV, or BLCA. On the other hand, for HDAC2-associated transcriptome profiles, we observed a significant depletion of immune-related terms, which complies with previous findings. Processes with ancestors related to cell cycle, cell division, and DNA repair were highly enriched for HDAC2 while depleted for HDAC7. These processes were previously reported as associated with stem cell features [11]. 

We performed additional validation through GSEA using curated gene sets from chemical and genetic perturbations collection (C2:CGP). We observed that HDAC2-related transcriptome profiles are significantly enriched across studied tumors with terms like DREAM targets, upregulation of epithelial–mesenchymal transition, G2/M cell cycle, E2F targets, EZH2 targets, and proliferation (Appendix A). In contrast, HDAC7-related transcriptome profiles are significantly depleted within these terms in the same tumors. In BRCA, LUAD, PRAD, and SARC, HDAC2—contrary to HDAC7—is associated with the enrichment of SOX2 and OCT4 targets. Conversely, the HDAC7 profile is enriched with immune-related terms like IL-4 signaling, differentiating of T cells, Th1 cell cytotoxicity, IFN-β targets, and IFN-γ response in HNSC and SARC. These observations support previous findings that HDAC2-associated transcriptome profiles are significantly enriched with stemness-related hallmarks of cancer, while HDAC7-associated transcriptome profiles are enriched with immune-related terms.

### 2.5. HDAC Expression Correlates with the Infiltration of Selected Immune Cell Subtypes, and HDAC7-High Expressing Tumors Exhibit Significant Upregulation of Distinct Chemoattractants

Next, we evaluated the tumor microenvironment concerning HDAC2 or HDAC7 expression in all tested tumor types. As presented in Figure 8A and Appendix A, the expression of HDAC2 correlates negatively with the leukocyte fraction [34] and immune checkpoint molecules in most tumors, while HDAC7 is mostly positively associated. 

To validate the observation of HDAC-dependent tumor microenvironment, we employed the ESTIMATE tool [35] and observed that the purity, immune, and stromal scores correlate highly positively with HDAC7 gene expression in selected solid tumors, except for OV and BLCA (Appendix A). On the other hand, we observed the opposite statistically significant correlation trend for HDAC2 expression in studied tumors, apart from LIHC. This confirms high immune infiltration and stroma presence in the tumor tissues associated with HDAC7-high and HDAC2-low expression. Low–high HDAC2/7 expression levels significantly differentiate the immune and stromal scores between patients in these groups (Appendix A). In other words, HDAC7-high and HDAC2-low patients have significantly higher tumor immune and stromal infiltration levels in many tumors.

To further examine immune infiltration, we used the TIMER2.0 database [36] and employed the xCELL algorithm of cell type quantification [37]. Compared to the expression of HDAC2 in many studied tumors, we observed a greater number of significant positive correlations between HDAC7 expression and different immune subpopulation levels (Figure 8B). This may suggest a strong positive relationship between HDAC7 and tumor infiltrating cells (TILs), especially considering CD4+ T cell populations whose levels generally correlate positively with HDAC7 expression and negatively with HDAC2 expression. Interestingly, myeloid-derived suppressor cells (whose level was evaluated through the TIDE algorithm [38]) strongly correlate with HDAC2 expression in all studied solid tumors, which is not observed for HDAC7. Furthermore, we looked at the levels of chemoattractants in all tested tumor types regarding HDAC2 (Appendix A) or HDAC7 (Figure 9) expression and observed that HDAC7 upregulation significantly associates with elevation of chemokines and their recognizing receptors. Moreover, HDAC7 expression is significantly associated with the promotion of inflammatory factors in all 11 tumors (Appendix A). In the case of HDAC2, we observed such an elevation in only BRCA, LIHC, and PRAD, while depletion in COAD and LUSC.

Finally, we compared the expression levels of HDAC2 and HDAC7 in studied tumors between patients with different transcriptome-based immune subtypes [34]. As shown in Appendix A, the median HDAC2 expression is generally lower in patients with immune subtypes C3 (inflammatory type) and C6 (TGF-b dominant type) relative to other subtypes in all tumors of interest besides (1) SARC, where the TGF-b dominant subtype features the highest relative expression of HDAC2 among all tumors, and (2) wherever no samples were classified as C6 subtype (OV, PRAD, READ, and LIHC). As shown in Appendix A, differences in HDAC7 expression between patients with different immune subtypes are statistically significant for all tumors (except OV and COAD) and *p*-values are generally lower in comparison to HDAC2 differences in the same cancers. Interestingly, C3 and C6 subtypes are associated with higher HDAC7 expression than other immune subtypes in LUSC and STAD.

### 2.6. HDAC2/7 as Potential Biomarkers of Immunotherapy Response in Solid Tumors

We searched the TIGER database [39] to explore the differential expression of class I and class IIA HDACs between responders and non-responders to immunotherapy in different tumors (Figure 10 and Appendix A). Among available studies, the only statistically significant difference between such patients was found in SKCM, where responders to combined anti-CTLA4+anti-PD1 therapy were characterized with lower HDAC2 expression than non-responders (Figure 10A). Although differences are not statistically significant for other available studies, preliminary yet no definitive results suggest the possible potential of HDAC2 and HDAC7 as immunotherapy response biomarkers, which can be confirmed only if more samples are available. In the case of HDAC2, this regards anti-PD1 therapy against KIRC and STAD (Figure 10B,E) and anti-CTLA-4 therapy against SKCM (Figure 10C). HDAC7 could eventually become a potential response biomarker in the case of KIRC anti-PD1 immunotherapy (Figure 10J). As patient cohorts in this analysis are too small, we underline that these findings are only suggestive, indicating the possible and promising direction of further research. Additional analyses should also be included to definitely prove the HDAC2/7 associations with immune response. In particular, HDAC2/7 activity and protein levels should be measured, alongside their chromatin accessibility.

## 3. Discussion

This is the first comprehensive report revealing the associations between HDAC family members, cancer stemness, and anti-tumor immune response across more than 20 types of solid tumors. Here, based on the transcriptomic, genomic, and clinical data from the TCGA database [24], Prognoscan platform [26], and TIGER database [39], and using several other bioinformatic tools [31,35,36,39,40,41], we demonstrated that the expression levels of class I and class II HDAC genes are related to cancer stemness and immunity, however, in a distinct manner. While class I HDAC family members (HDAC1/2/3/8) were positively associated with cancer stemness across most of the tested tumor types, class IIA HDAC members (HDAC4/5/7/9) exhibited an inverse correlation. In support of these observations, we found that the most robust and stringent positive associations with cancer stemness were evidenced for HDAC2 expression, whereas a negative association was demonstrated for HDAC7 expression. Importantly, our data clearly highlighted the enrichment of stemness-associated “Hallmarks” terms in HDAC2-related transcriptome profiles and significant depletion of those terms in HDAC7-high expressing tumors. Furthermore, our analyses demonstrated that high HDAC2 levels are linked to the immune cell infiltration status resembling the immunosuppressive environment. In contrast, HDAC7 expression level positively correlates with increased immune cells’ abundance, suggesting an augmented cancer immunogenicity. In line with these data, we noted a very strong and versatile positive association between HDAC7 level and the expression of chemokines or chemokine receptors. Finally, our analyses suggest that HDAC2 expression may have a potential value as a biomarker of immunotherapy.

The involvement of HDAC family members in cancerogenesis has been studied for several decades, identifying specific HDACs’ roles in distinct cellular and molecular events, including cell cycle and cellular proliferation, apoptosis, DNA damage response, autophagy, EMT, or angiogenesis [42,43]—all crucial players in cancer development and progression. Also, the total pan-HDAC activity, including all family members, was determined as essential for proper cell differentiation and regulation of pluripotency of normal stem cells [44,45,46]. However, the role of specific HDAC members in facilitating the acquisition or maintenance of cancer stem cell-like phenotype has not been fully explored. Also, the involvement of most HDAC members in regulating anti-tumor immune response remains elusive.

Here, using transcriptomic, genomic, and clinical data for diverse solid tumors, we investigated the association of HDAC expression, cancer stemness, and anti-tumor immunity. Previous studies demonstrated the overexpression of class I HDAC members in different cancer types, including gastric, esophagus, colorectal, prostate, glioma, melanoma, lung, and breast cancers (reviewed in [42]). Also, their upregulation is frequently associated with poor prognosis, especially in lung [47], gastric [48], liver [49], colorectal [50], ovarian [51], bladder [52], and breast carcinomas [53]. Here, we show a consistent upregulation of all class I HDACs in tumor tissues in contrast to other classes, especially IIA and IIB HDAC members, whose expression pattern is more divergent and tumor type specific. Using clinical data from the TCGA and Prognoscan databases, we demonstrated that high class I HDAC members’ expression corresponds with worse survival of patients, especially in BRCA, LUAD, and GBM. Our results are in line with the previously reported worse survival rate of cancer patients with overexpressing class I HDACs, especially HDAC1 and HDAC3 in lung cancers [47], HDAC2 in liver [49] or breast cancers [54], and HDAC1/2/3 in ovarian cancers [55], gastric cancers [56], or sarcomas [57]. 

In comparison, high class IIA HDAC members’ expression is associated with better survival of BRCA, COAD, LUAD, and GBM patients. Our results align with a previously demonstrated better outcome for high-expressing patients of specific class IIA HDAC members, specifically in non-small cell lung carcinomas [58], HDAC7 in triple-negative breast cancers [59], and HDAC4/5 in gliomas [60]. 

Also, HDACs are differentially associated with tumor stage and grade, however, in a cancer-dependent and gene-specific manner, and do not follow any specific trends, even within each HDAC family class. We observed several distinct HDACs associated with tumor grade in LGG, KIRC, and BLCA, while significant associations with stage are primarily observed in KIRC, KIRP, BLCA, and TGCT. Previously, higher expression of class I HDACs was observed in higher stages of colon cancer [61]. In childhood neuroblastoma, upregulation of HDAC8 was associated with advanced stage, poor prognosis, and poor survival [62]. Elevated expression of class I HDACs was also shown in higher-grade prostate cancers [63]. Higher-grade ovarian tumors are characterized by upregulation of HDAC2 [64]. Also, HDAC2 overexpression was associated with more aggressive stage III breast cancers, which significantly correlated with a worse prognosis [65]. HDAC2 overexpression showed a statistically significant correlation with increased lymphatic spreading of the tumor (N stage) and lower tumor differentiation (higher grade) in esophageal adenocarcinomas [66]. In lung cancer, HDAC1 levels were substantially lower in patients with well-differentiated adenocarcinoma than in those with a lower differentiation grade [47]. Both HDAC1 and HDAC2 were significantly associated with higher tumor grades of urothelial bladder carcinoma [52]. Together, highly expressed class I HDACs are usually associated with terminal illness and inferior outcomes in cancer patients.

Class II HDACs (HDAC5/7/9) downregulation was observed in glioblastomas compared to grade I–II astrocytomas [67]. Higher-grade ovarian tumors exhibit downregulation of HDAC4 [64]. Li H. et al. [68] demonstrated that HDAC9 expression is negatively associated with both T and N stages (albeit not correlated with clinical stages) in PDAC tissues. Also, HDAC6 expression was significantly associated with earlier histopathological stages of pancreatic adenocarcinoma [69]. Taken together, class II HDACs are usually associated with less progressed disease and superior outcomes in cancer patients.

Next, we demonstrated very robust and positive associations of class I HDACs’ expression and cancer stemness followed by a negative association of class IIA HDACs levels and tumor dedifferentiation status as measured with previously reported stemness quantifiers: the mRNA-stemness index (mRNA-SI) and other stem cell-derived gene signatures: Ben-Porath ES2, Wong ESC, and Ben-Porath ES core [11,27,28]. The mRNA-SI was developed by a one-class logistic regression algorithm on transcriptomic data extracted from distinct stem cell populations and their differentiated progeny, creating a comprehensive stemness signature that allows for the quantification of tumor dedifferentiation in almost 12,000 samples across 33 tumor types. 

Ben-Porath I. et al. [28] have found that histologically poorly differentiated tumors show preferential overexpression of genes typically enriched in embryonic stem (ES) cells. They demonstrated that this ES-like signature was associated with high-grade estrogen receptor (ER)-negative tumors, often of the basal-like subtype, and with poor clinical outcomes. Moreover, the ES signature was present in poorly differentiated glioblastomas and bladder carcinomas, suggesting its versatility in the acquisition of cancer dedifferentiation status regardless of the tumor type. Furthermore, Wong DJ. et al. [27] have recognized the embryonic stem cell (ESC) transcriptional program that is frequently activated in diverse human epithelial cancers and strongly predicts metastasis and death. They also identified that the c-Myc oncogene is sufficient to reactivate cancer cells’ ESC-like program. Using these independent transcriptome signatures, we quantified the level of cancer stemness in diverse solid tumors. We demonstrated that class I HDAC members’ expression was positively associated with the strongest correlations for HDAC2 regardless of the stemness score or signature. In contrast, the level of class IIA HDACs correlates negatively with the most robust and consistent associations observed for HDAC7.

Previously, Saunders A. et al. [70] demonstrated that HDAC2 is critical for the reprogramming-promoting function of the SIN3A complex, facilitating the acquisition of pluripotency by non-transformed cells in NANOG-driven reprogramming. The SIN3A/HDAC2 complex and NANOG transcription factor are required to directly induce a synergistic transcriptional program, encompassing the activation of pluripotency genes and repression of differentiating genes. In light of cancer stemness acquisition significantly resembling somatic cell reprogramming, the abovementioned results strongly suggest that HDAC2 could contribute to the stem cell-like properties of cancer cells. Furthermore, both HDAC1 and HDAC2 were identified as proteins present in complexes with SOX2 transcription factor [71], further supporting the stem cell-associated roles for those class I HDACs. Also, the oncogenic activity of MYC is significantly induced by HDAC2, suggesting the potential benefits of applying HDAC inhibitors in the prevention and treatment of Myc-driven cancers. Recently, Bahia RK. et al. [72] have identified HDAC2 as the most relevant histone deacetylase that facilitates stem cell-like properties in brain tumor cells. HDAC2 activity regulates chromatin compaction that impacts the expression of SMAD3 and SOX2 and is, thus, critical for the self-renewal of brain cancer cells. Also, the inhibition of HDAC2 activity disrupted the interaction with SMAD3, resulting in the loss of stem cell-like characteristics of brain tumor cells. 

In our data, class I HDAC expression, especially HDAC2, correlates with the level of selected pluripotency markers, including BMI1, OCT4, MYC, and SOX2. Also, the associations were the strongest in testicular germ cell tumors, which exhibit the most pronounced stem cell-like phenotype of all tested solid tumors. On the other hand, class II HDACs, particularly HDAC7, are associated negatively with pluripotency markers’ expression in most tumor types, except for OCT4 (encoded by POU5F1), and NANOG in KIRC, KIRP, LGG, LUAD, LUSC, PRAD, and THCA. This suggests that class I HDACs may occur as attractive targets for anti-cancer treatment, especially for dedifferentiated (stem cell-like) solid tumors. 

When defining their transcriptome-based stemness score, Malta T. et al. [11] have revealed an unanticipated negative correlation of cancer stemness with immune checkpoint expression and infiltrating immune cells. This was further supported by Miranda A. et al. [13], who observed that cancer stemness is associated with suppressed immune response, higher intratumoral heterogeneity, and dramatically worse outcomes for most TCGA cancers. The persistent interaction of cancer stem cells with the tumor microenvironment (TME) provides the ability to avoid recognition and elimination by immune cells, facilitating CSC’s survival and tumor progression [73]. Cancer stem cells protect themselves against immune surveillance through several distinct mechanisms, including suppression of T cell activation, aberrant MHC class I expression, repression of tumor-associated antigens (TAAs), or by exploiting the immunosuppressive function of multiple immune checkpoint (IC) molecules [74]. As cancer stemness maintenance relies significantly on epigenetic mechanisms, restoring dysregulated histone modifications to overcome cancer resistance to immunotherapy is a promising anticancer strategy.

Here, we demonstrated that HDAC2-high expressing tumors that exhibit enriched stem cell-like phenotypes are significantly depleted with immune cells, while the remaining infiltrating populations are responsible for the formation of an immunosuppressive microenvironment. On the other hand, solid tumors with HDAC7 upregulation are enriched with specific T cell subpopulations, including CD4+ naive and central memory T cells and CD8+ T cells, and activated dendritic cells, suggesting the formation of T cell-inflamed tumors [75]. HDAC7 overexpression is associated with the upregulation of vast chemokine molecules, including the following T cell-attracting chemokines [75,76]: CCL2, CCL3, CCL4, CCL5, CXCL9, and CXCL10, which explains their strong immune cell infiltration. These results further correspond with a significant upregulation of HDAC7 and a robust depletion of HDAC2 in the C3 immune subtype (inflammatory) in most tested tumors. However, HDAC7 is significantly associated with the upregulation of immune checkpoint molecules, including PD-1, PD-L1, and CTLA4. These suggest that, despite HDAC7-high expressing tumors are heavily infiltrated with immune cells that could kill cancer cells, the effectiveness of immune responses might be significantly abrogated at the level of immune checkpoint signaling. Therefore, HDAC inhibitors represent an exciting opportunity to improve the efficacy of immunotherapeutic regimens. Mechanistically, the tumor microenvironment and specific anti-tumor immune responses might be affected by HDAC inhibitors at several distinct levels, including the stimulation of cancer antigens’ expression or MHC class I/II expression, the modulation of immunosuppressive signaling pathways, the reduction in immunosuppressive cell populations, or the enrichment of chemokine expression [77,78]. 

Previously, Wang H-F. et al. have shown significant elimination of MDSC in the microenvironment of mice breast tumors treated with HDAC inhibitor SAHA (suberoylanilide hydroxamic acid, a potent inhibitor of HDAC1/2/3/6/7/11) that corresponds with an increased proportion of T cells (particularly that of IFN-γ- or perforin-producing CD8+ T cells) [79]. Yan M. et al. [53] have demonstrated that class I HDAC members associate negatively with CD8+ effector T cells, NK, and NKT cells in gynecologic cancers, including BRCA, CESC, OV, and UCEC tumors. Also, class I HDAC expression significantly correlated with the downregulation of specific T cell marker genes and suppressed inflammatory markers. In their study, the combination of HDAC inhibitor SAHA with anti-PD-1 in breast tumor-bearing mice suppressed tumor cell proliferation, promoted inflammatory responses, and increased the numbers of tumor-infiltrating T lymphocytes in vivo. Furthermore, Yang et al. [80] observed that selective HDAC8 inhibition resulted in increased CD8+ T cell tumor infiltration in a preclinical model of hepatocellular carcinoma due to elevated production of T cell-recruiting chemokines. Direct inhibition of HDAC8 coupled with PD-L1 blockade further reinvigorated CD8+ T cells, turning from a functional exhausted state to IL-2- and IFN-γ-producing TILs.

Nowadays, the involvement of epigenetic dysregulation in accelerating cancer progression (at least partially by facilitating cancer stemness) is unquestionable. An increasing number of studies demonstrate the potential of HDAC family members to become druggable targets in solid tumors, albeit those studies do not raise the question of therapeutic targeting of the stem cell-like compartment. Also, the fact that specific HDACs are essential players in molecular mechanisms modulating cancer stemness is mainly ignored. Taking that into consideration and based on our results, we suggest that direct inhibition of class I HDAC family members (which are significantly overexpressed in stem cell-like, low-infiltrated solid tumors), together with immune checkpoint inhibitors, might result in a better outcome of treated patients, presumably by abolishing the self-renewal properties of cancer cells and by strengthening the anti-tumor immune responses. Our observations stay in line with previously reported enhanced antitumor immunity in triple-negative breast cancers achieved by HDAC2 knockout in the breast cancer mice models. As presented by Xu P. et al. [81], HDAC2 is required for the chromatin remodeling of IFNγ-induced PD-L1 expression in breast tumors, and direct HDAC2 targeting suppresses immune escape of the tumor. Also, Zheng H. et al. [82] have found that class I HDAC inhibitor romidepsin induced a strong T cell-dependent antitumor response and enhanced the therapeutic effect of PD-1 inhibitors in lung adenocarcinoma. Later on, Orillion A. et al. [83] reported the enhancement of the antitumor effect of PD-1 inhibition by entinostat, another class I HDAC inhibitor, in murine models of lung and renal cell carcinomas. Recently, Han R. et al. [84] have elegantly summarized the rationale of targeting HDAC2 and immune checkpoint inhibitors in hepatocellular carcinomas. This novel tumor treatment strategy is endowed with great clinical application and research prospects, which provides a new opportunity to improve the overall prognosis of cancer patients further. Therefore, we suggest that HDAC2-high expressing cancer patients who exhibit stem cell-like tumor phenotype might benefit from the combination therapy, including both class I HDAC and immune checkpoint inhibitors.

## 4. Materials and Methods

### 4.1. TCGA Solid Tumor Types Selected for the Study

In the current study, we initially selected for the analysis 22 solid TCGA [24,85] tumor types with more than 100 mRNA-SeqV2 available samples (Table 2). Samples within individual studies (namely, tumor types) included in the TCGA come from primary untreated tumor resection fragments which are composed of at least 80% tumor nuclei [85]. All data (including raw mRNA bulk sequencing results and clinical information) are available online, and the access is unrestricted and does not require patients’ consent or other permissions. The use of the data does not violate any personal or institutional rights.

### 4.2. TCGA Genetic and Clinical Data

The RNA sequencing-based mRNA expression data were directly downloaded from the cBioportal [86] through web-API (https://www.cbioportal.org/webAPI, accessed on 15 October 2023). RNASeq V2 from TCGA is processed and normalized using RSEM [87]. Specifically, the RNASeq V2 data in cBioPortal corresponds to the rsem.genes.normalized_results file from TCGA. Expression data of HDAC family members in cancer and normal adjacent samples for pan-can overviews were taken through the UCSC Xena Browser [88] from GDC [89] and GTEx [25] datasets, respectively (https://xena.ucsc.edu/, accessed on 22 October 2023). Specific chemokines and chemokine receptors associated with HDAC2/7 expression were selected through the TISIDB (Tumor-Immune System Interaction Database) web-based resource (http://cis.hku.hk/TISIDB/index.php, accessed on 9 May 2024) [41]. Expression data of chemokines, chemokine receptors, literature-specific inflammatory genes, transcription factors, and immune checkpoint genes in selected cancers was directly downloaded from the TCGA dataset using the UCSC Xena Browser (https://xena.ucsc.edu/, accessed on 9 May 2024) [88]. All clinical data (including grade, stage, tumor detailed subtype, tumor size, lymph nodes, and metastasis) for each sample was downloaded directly from the cBioPortal. Survival analysis (OS and DFS) was conducted with the GEPIA2 database (http://gepia2.cancer-pku.cn/, accessed on 28 October 2023) [90]. The hazard ratio was estimated through the Mantel–Cox test for patients with high (Q3, 75th percentile) relative to low (Q1, 25th percentile) expression of specific HDAC family members. Statistical values were extracted from resulting individual plots using the Selenium WebDriver [91]. 

### 4.3. Prognosis Analysis Using the Prognoscan Database

The PrognoScan (http://dna00.bio.kyutech.ac.jp/PrognoScan/, accessed on 8 May 2024) [26] database was used for the meta-analysis of the prognostic value of various genes. This online platform assists in investigating the relationship between gene expression and patient prognosis across a large collection of cancer microarray datasets. The significance threshold of associations between HDAC family members’ expression and patients’ overall survival was adjusted to a Cox *p*-value < 0.05. 

### 4.4. Stemness-Associated Scores

The mRNA-SI stemness score and other stemness signatures (Ben-Porath ES core, Ben-Porath ES2, and Wong ESC_core) used in this study were previously described [11,27,28]. Briefly, the mRNA-SI signature was calculated based on a previously built predictive model using one-class logistic regression (OCLR) on the pluripotent stem cell samples (ESC and iPSC) from the Progenitor Cell Biology Consortium (PCBC) dataset. The obtained signature was further applied to score TCGA samples using the Spearman correlations between the model’s weight vector and the sample’s expression profile. The index was subsequently mapped to the [0, 1] range. As for Ben-Porat ES core, Ben-Porath ES2, and Wong ESC core signatures, we used the Gene Set Cancer Analysis (GSCA) platform (https://guolab.wchscu.cn/GSCA/, accessed on 20 November 2023) to calculate the enrichment score of inputted gene sets in each sample of selected cancers with Gene Set Variation Analysis (GSVA) method [40,92].

### 4.5. Gene Set Enrichment Analysis

We employed the Enrichr tool (https://maayanlab.cloud/Enrichr/, accessed on 13 January 2024) which is an integrative web-based software application providing various types of computing gene set enrichment, and visualization summaries of collective functions of single genes or gene lists [33]. We used the top 100 most relevant genes (identified with ARCHS4 RNA-seq gene–gene co-expression matrix) for a queried gene (HDAC2 or HDAC7) to determine significantly enriched pathways (MSigDB Hallmark 2020 module) or to later on detect potential targets for known transcription factors (Transcription Factor PPIs module).

The Gene Set Enrichment Analysis (GSEA, https://www.gsea-msigdb.org/, accessed on 16 February 2024) [31] was performed to detect coordinated expression of a priori-defined groups of genes within the tested samples. Gene sets are available at the Molecular Signatures Database (https://www.gsea-msigdb.org/gsea/msigdb/index.jsp, accessed on 16 February 2024) [93]. All significantly differentially expressed genes were previously ranked based on their log2FC between analyzed groups: high expression (Q3, 75th percentile) relative to low (Q1, 25th percentile) expression of HDAC2 or HDAC7. Groups of ranked genes were imported to GSEA, and the GSEAPreranked tool was run according to the following parameters: Dataset used in the original format (no collapse) and permutation number = 1000. The FDR <0.05 was used to correct for multiple comparisons and gene set sizes. For the CP5:BP collection, the resulting lists of terms for separately HDAC2 and HDAC7 genes were limited to processes for which data are available for all studied tumor types. We selected processes from the very top and bottom (measured as sum of NESs in each tumor) of the pre-ranked datasets. Both lists were intersected to obtain a final list for both genes combined. Resulting processes were manually annotated in an unbiased manner, based on the most relevant ancestry processes (retrieved from AmiGO2 resource [94], https://amigo.geneontology.org/, accessed on 20 February 2024). This step resulted in classification into cell cycle, cell division, DNA repair, and immune process classes. For the CP2:CGP collection, we manually filtered out all terms which are specific and thus applicable only for individual tumor types and intersected the rest of processes for both HDAC2 and HDAC7 genes. The hallmarks collection is presented without any prior filtering.

### 4.6. Immune and Stromal Infiltration

Immune infiltration was examined using the TIMER2.0 database [36], which allowed us to calculate correlations of gene expressions with immune infiltration levels in diverse cancer types using different algorithms. The purity-adjusted Spearman’s rho values across 11 cancer types calculated using the xCELL method [37] were downloaded directly from TIMER2.0 (http://timer.cistrome.org, accessed 14 February 2024) [36]. Based on single-sample GSEA, xCELL is a cell-type quantification method that, unlike other commonly used algorithms like CIBERSORT, is a gene-based marker approach rather than a deconvolutional approach. It outperforms other methods, allows predicting the highest number of cell types among available algorithms (up to 64), and is considered more robust for abundance analysis in contrast to deconvolutional methods [95].

ESTIMATE (estimation of stromal and immune cells in malignant tumor tissues using expression data) web-based software was employed to predict each tumor purity, level of stroma cells presence in tumor tissues, and level of tumor immune infiltration, using gene expression data (https://bioinformatics.mdanderson.org/estimate/, accessed on 5 February 2024) [35].

### 4.7. Immune Subtypes and Immunotherapy Results

To investigate tumor–immune interactions, we used the TISIDB (tumor–immune system interaction database) web-based resource (http://cis.hku.hk/TISIDB/index.php, accessed on 20 February 2024) [41]. We explored relationships between HDAC2 or HDAC7 and five immune subtypes: C1 (wound healing), C2 (IFN-gamma dominant), C3 (inflammatory), C4 (lymphocyte depleted), C6 (TGF-b dominant) in 11 cancer types. 

Additionally, the TIGER database (http://tiger.canceromics.org/, accessed on 18 January 2024) [39] was employed to explore the potential role of HDAC family members as biomarkers of immunotherapy response in solid tumors. The immunotherapy response module provides differential expression analysis, which uses bulk transcriptome data with immunotherapy clinical information to find differences in the expression of the gene in question between responders and non-responders to specific immunotherapies [35].

### 4.8. Statistical Analysis

Statistical analyses were carried out with GraphPad Prism 8.0 (GraphPad Software, Inc., La Jolla, CA, USA), R 4.2.2 (R Foundation for Statistical Computing, Vienna, Austria) with ggplot2 [96] and complexheatmap [97] libraries for visualization, and Python 3.11 (The Python Software Foundation, Wilmington, Delaware) with Selenium [91] library for data retrieval. Exact applied statistical tests are described in each figure description. 

## 5. Conclusions

Our research uncovered a significant association between cancer stemness and an elevated expression of class I HDAC family members, especially HDAC2, where the association was robust and universal regardless of the tested tumor type. On the other hand, the relation of class IIA HDAC members is significantly negative, with HDAC7 exhibiting the strongest ones. We further demonstrated for the first time the distinct trends of associations between class I and class IIA HDACs and anti-tumor immunity, with the expression of class I HDAC2 being negatively correlated and class IIA HDAC7 being positively correlated. 

We suggest that patients with stem cell-like, low-infiltrated solid tumors exhibiting significant upregulation of HDAC2 might benefit from the treatment with the combination of HDAC2-specific inhibitors and immunotherapy (i.e., immune checkpoint inhibitors). 

## Figures and Tables

**Figure 1 ijms-25-07841-f001:**
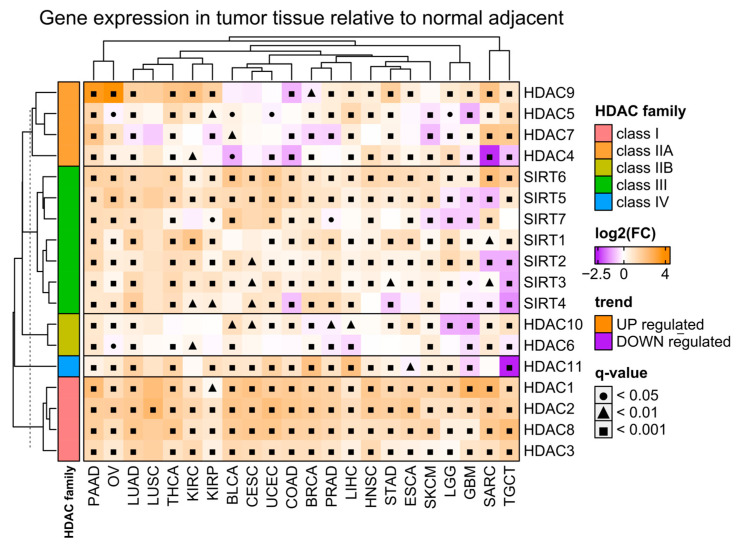
Differences in expression of HDAC family members in tumor tissues relative to normal adjacent tissues. Color on the heatmap denotes either upregulated (orange) or downregulated (purple) expression in tumor tissues. Welch’s *t*-test. Benjamin–Hochberg was used for multiple testing corrections. Tumor abbreviations are explained in Table 2.

**Figure 2 ijms-25-07841-f002:**
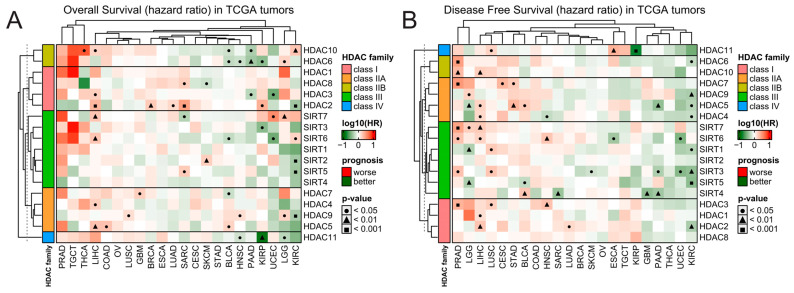
Patients’ survival analysis related to the expression of HDAC family genes across TCGA tumors. Hazard ratios (log10[HR]) of death for patients with high (Q3, 75th percentile) relative to low (Q1, 25th percentile) expression of specific HDAC family members for (**A**) overall survival and (**B**) disease-free survival. Red and green denote higher or lower hazard ratios, respectively, for the patients with a high expression of a given HDAC. Benjamin–Hochberg was used for multiple testing corrections.

**Figure 3 ijms-25-07841-f003:**
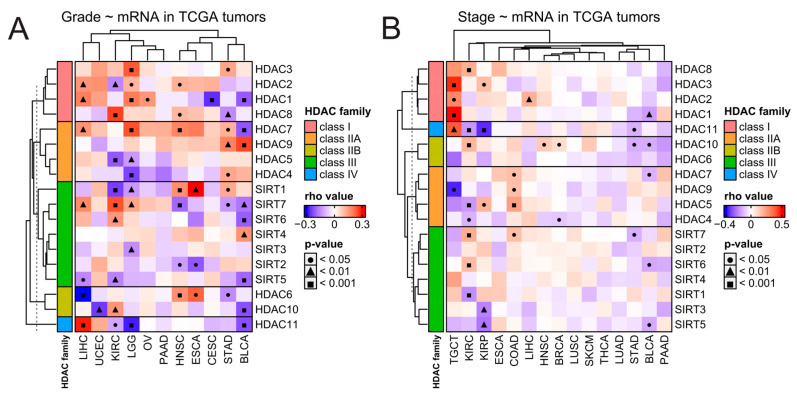
Clinical overview of HDAC genes in solid tumors. Correlations between expression of HDAC family genes and tumors’ (**A**) grading and (**B**) staging. Spearman’s test with asymptotic *t*-test for *p*-values. Benjamin–Hochberg was used for multiple testing corrections. COAD stands here as a double abbreviation for COADREAD.

**Figure 4 ijms-25-07841-f004:**
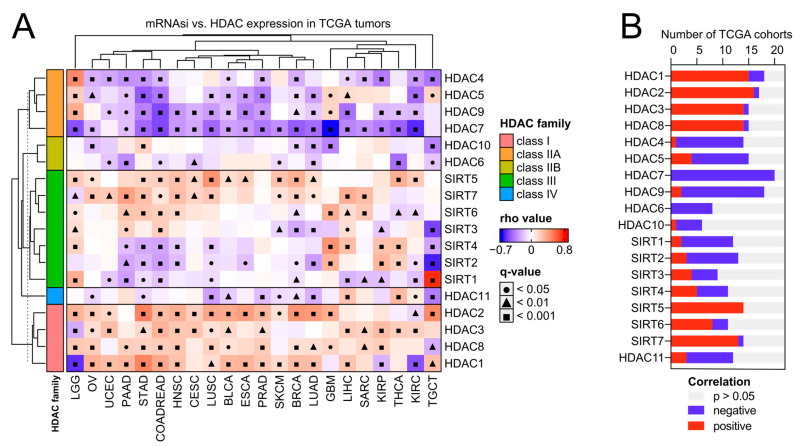
Class I HDACs and class IIA HDACs represent universal and opposite patterns of associations with cancer stemness across distinct types of solid tumors. (**A**) Correlations between the expression of HDAC family genes and mRNA stemness index (mRNAsi) across 22 TCGA solid tumors. Spearman’s test with asymptotic *t*-test for *p*-values. Benjamin–Hochberg was used for multiple testing corrections. (**B**) The number of TCGA cohorts that exhibit either positive (red) or negative (blue) correlation between the expression of specific HDAC family members and tumor dedifferentiation status defined by mRNAsi. The number of statistically insignificant associations is marked with gray.

**Figure 5 ijms-25-07841-f005:**
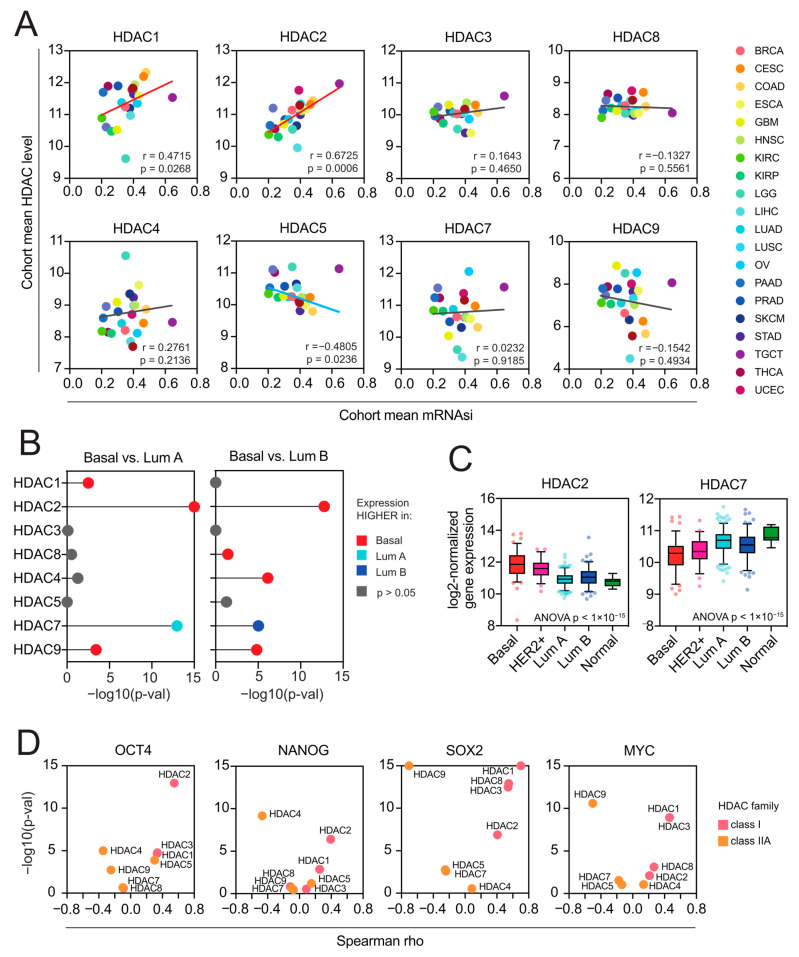
Class I HDACs are positively associated with cancer stemness, while class IIA HDACs are negatively associated with cancer stemness. (**A**) The association between the cohort mean mRNA-SI level and the cohort mean expression (log2-normalized) of class I and class IIA HDAC genes across 22 TCGA cohorts. Each cancer type is color coded in the dot plots. (**B**) Class I and class IIA HDAC genes are differentially expressed between molecular (PAM50) BRCA subtypes. Statistical significance (−log10(adj. *p*-value)) of comparisons between basal vs. luminal A and basal vs. luminal B BRCA subtypes calculated with the Kruskal–Wallis test followed by Dunn’s multiple comparisons test are denoted in the lollipop plots and color coded accordingly: red—HDAC expression higher in basal vs. luminal A or basal vs. luminal B; light blue—HDAC expression lower in basal vs. luminal A; dark blue—HDAC expression lower in basal vs. luminal B (B < Lb); gray—no statistical significance. (**C**) The expression of HDAC2 and HDAC7 genes in TCGA BRCA samples stratified by molecular subtypes (PAM50) into 5 subgroups: basal (red), HER2-positive (magenta), luminal A (dark blue), luminal B (light blue), and normal-like (green). The mean value with standard deviation (SD) is plotted. (**D**) Class I HDAC members correlate positively, and class IIA HDAC genes correlate negatively with the expression of several well-known pluripotency markers (OCT4, NANOG, SOX2, MYC) in stem cell-like testicular germ cell tumors (TGCT, *n* = 156). Spearman correlation coefficient and statistical significance (−log10(*p*-val)) is denoted.

**Figure 6 ijms-25-07841-f006:**
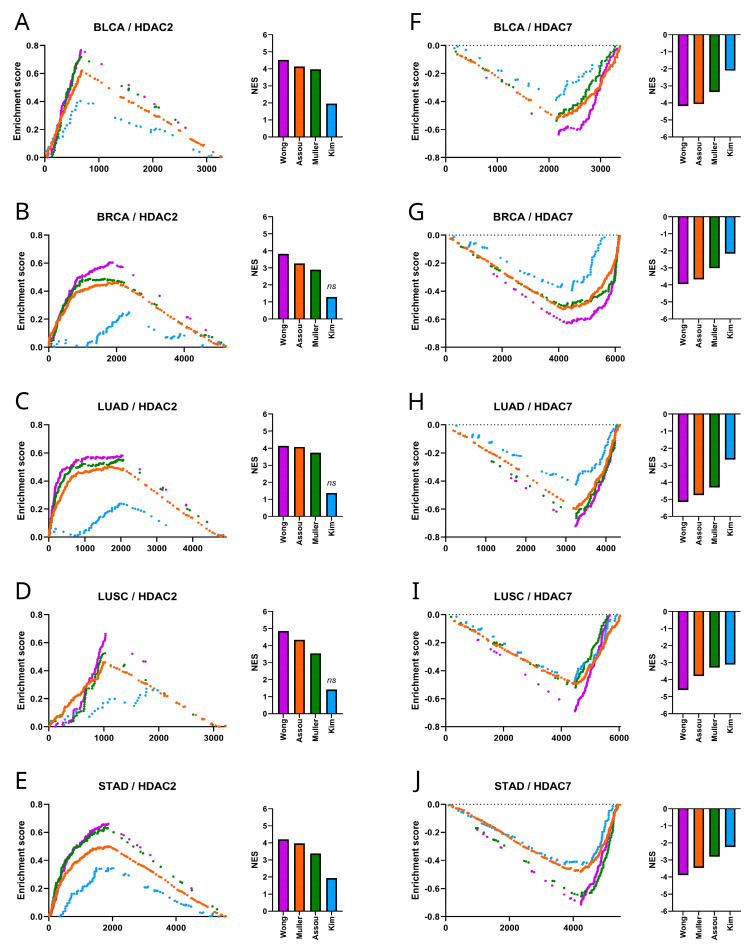
Stemness gene signatures in HDAC2- and HDAC7-associated transcriptome profiles. (**A**–**E**) The GSEA of stemness gene sets (Muller Plurinet, Wong ESC, Kim Myc, Assou ESC) correlated to HDAC2 revealed significant enrichment for most gene sets (*p* < 0.01) (not including Kim_Myc) in (**A**) BLCA, (**B**) BRCA, (**C**) LUAD, (**D**) LUSC, and (**E**) STAD. The normalized enrichment score (NES) for each gene signature is plotted in the bar graph. (**F**–**J**) Similarly, the GSEA of the same gene sets correlated to HDAC7 showed significant depletion (*p* < 0.0001) in (**F**) BLCA, (**G**) BRCA, (**H**) LUAD, (**I**) LUSC, and (**J**) STAD. The normalized enrichment score (NES) for each gene signature is plotted in the bar graph; *ns*—not significant.

**Figure 7 ijms-25-07841-f007:**
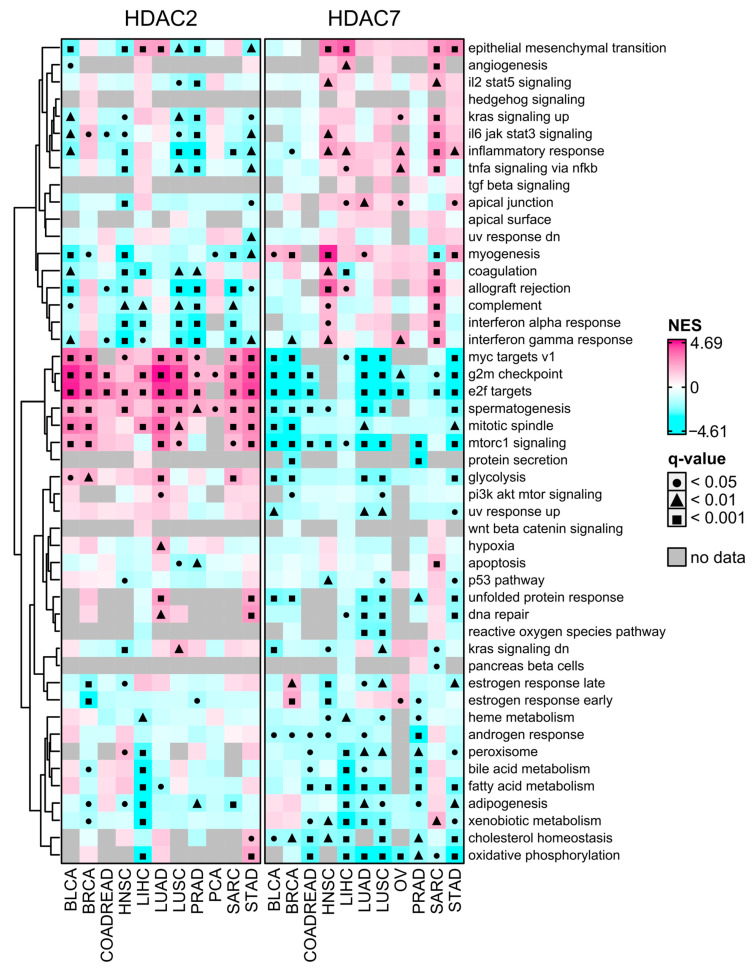
MSigDB Hallmark Gene Set Enrichment (GSEA) preranked analysis of HDAC2 and HDAC7 gene expression across selected TCGA studies. The high-expression group (Q3, 75th percentile) relative to the low-expression group (Q1, 25th percentile). Negative and positive NES values indicate the enrichment of selected hallmark gene sets at the bottom or the top of the ranked dataset. Gray cells indicate no available data. Benjamin–Hochberg was used for multiple testing corrections.

**Figure 8 ijms-25-07841-f008:**
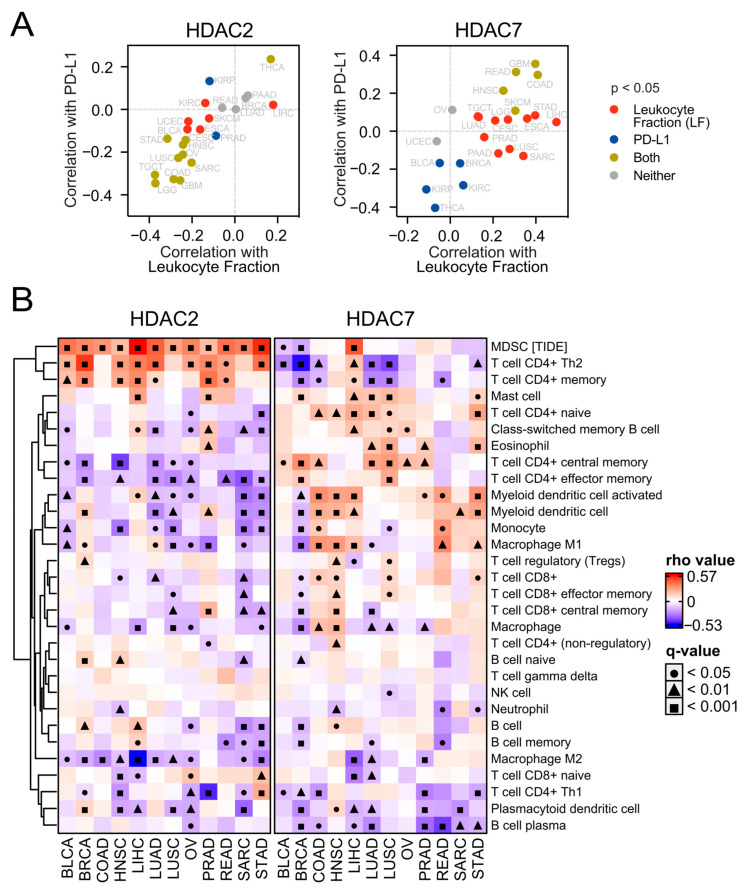
HDAC2 and HDAC7 expression levels exhibit opposite associations with the tumor microenvironment. (**A**) HDAC2 and HDAC7 expression in the context of tumor microenvironment. Each panel shows the Spearman correlation between the HDAC2 (left) or HDAC7 (right) level and PD-L1 protein expression plotted against the Spearman correlation between the same HDAC and total leukocyte fraction (LF). (**B**) Correlations of HDAC2 and HDAC7 gene expression with infiltrating immune subpopulation levels across selected TCGA studies. The analysis is based on the XCELL algorithm (in the TIMER2.0 database), besides MDSC (myeloid-derived suppressor cells), which is based on the TIDE algorithm. Gray cells indicate no available data. Benjamin–Hochberg was used for multiple testing corrections.

**Figure 9 ijms-25-07841-f009:**
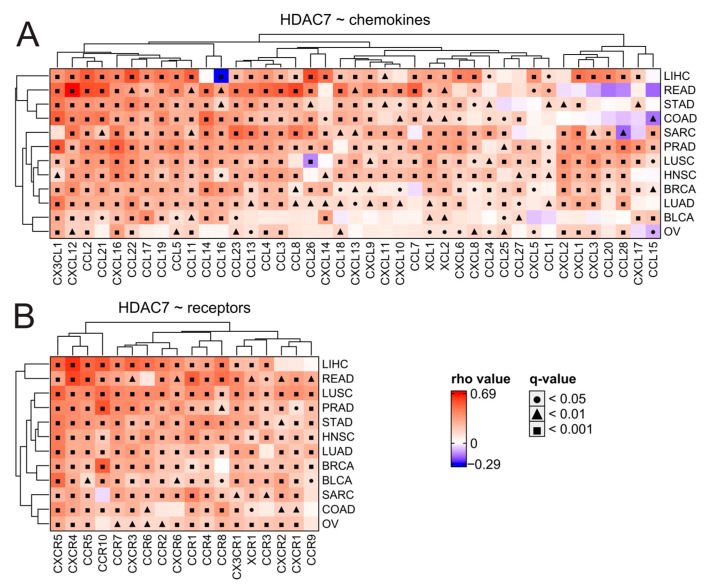
HDAC7 expression is associated significantly with the elevation of chemokines and their respective receptors in tested solid tumors. (**A**,**B**) Expression correlations of HDAC7 with (**A**) chemokines and (**B**) chemokine receptors across distinct tumor types. Spearman’s test with asymptotic *t*-test for *p*-values. Benjamin–Hochberg was used for multiple testing corrections.

**Figure 10 ijms-25-07841-f010:**
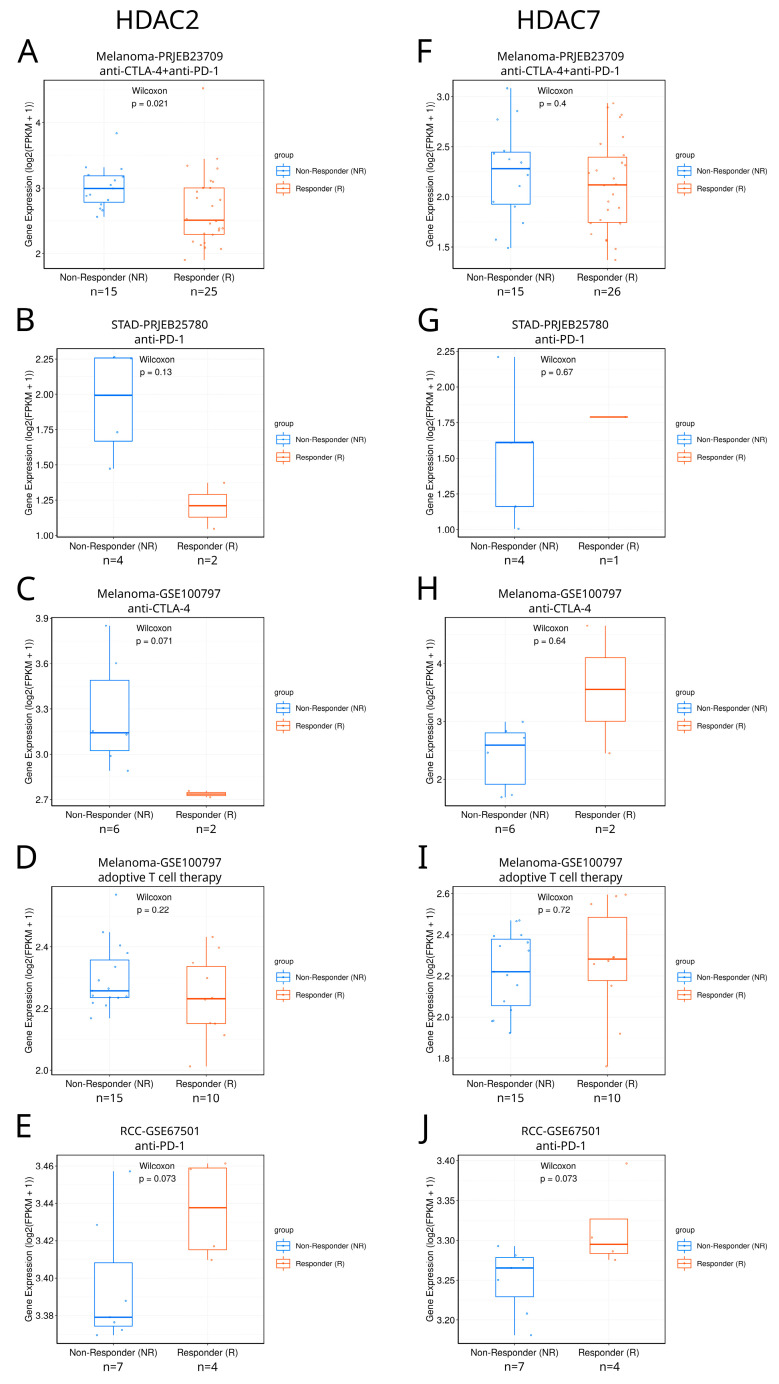
Differential expression of HDAC2 and HDAC7 between responders and non-responders to immunotherapy in different tumors. (**A**–**E**) HDAC2 expression in responders and non-responders in different studies: (**A**) Melanoma-PRJEB23709, (**B**) STAD-PRJEB25780, (**C**) Melanoma-GSE100797, (**D**) Melanoma-GSE100797, and (**E**) RCC-GSE67501 (renal cell carcinoma). (**F**–**J**) HDAC7 expression in responders and non-responders in different studies: (**F**) Melanoma-PRJEB23709, (**G**) STAD-PRJEB25780, (**H**) Melanoma-GSE100797, (**I**) Melanoma-GSE100797, and (**J**) RCC-GSE67501. Panels were taken directly from the TIGER database.

**Table 1 ijms-25-07841-t001:** Histone deacetylase superfamily hierarchy [23].

Subgroup	Group Genes
HDAC class I	HDAC1, HDAC2, HDAC3, HDAC8
HDAC class IIA	HDAC4, HDAC5, HDAC7, HDAC9
HDAC class IIB	HDAC6, HDAC10
HDAC class III/sirtuins	SIRT1, SIRT2, SIRT3, SIRT4, SIRT5, SIRT6, SIRT7
HDAC class IV	HDAC11

**Table 2 ijms-25-07841-t002:** Tumor abbreviations of TCGA studies used in the analysis.

Study Abbreviation	Study Name	N Samples [TCGA]
BLCA	Bladder urothelial carcinoma	408
BRCA	Breast invasive carcinoma	1100
CESC	Cervical squamous cell carcinoma and endocervical adenocarcinoma	306
COAD	Colon adenocarcinoma	470 ^1^
COADREAD	Colon and rectum adenocarcinoma	382
ESCA	Esophageal carcinoma	185
GBM	Glioblastoma multiforme	166
HNSC	Head and neck squamous cell carcinoma	522
KIRC	Kidney renal clear cell carcinoma	534
KIRP	Kidney renal papillary cell carcinoma	291
LGG	Low-grade glioma	530
LIHC	Liver hepatocellular carcinoma	373
LUAD	Lung adenocarcinoma	517
LUSC	Lung squamous cell carcinoma	501
OV	Ovarian serous cystadenocarcinoma	307
PAAD	Pancreatic adenocarcinoma	179
PRAD	Prostate adenocarcinoma	498
READ	Rectum adenocarcinoma	92 ^1^
SARC	Sarcoma	263
SKCM	Skin cutaneous melanoma	472
STAD	Stomach adenocarcinoma	415
TGCT	Testicular germ cell tumors	156
THCA	Thyroid carcinoma	509
UCEC	Uterine corpus endometrial carcinoma	177

^1^ Accessed from the XENA Browser, separate COAD and READ samples are not directly available through the cBioportal database.

## Data Availability

The datasets supporting the conclusions of this article are available in the TCGA and Prognoscan repositories—TCGA: BLCA, BRCA, CESC, ESCA, GBM, HNSC, KIRC, KIRC, LIHC, LUAD, LUSC, OV, PAAD, PRAD, SARC, SKCM, STAD, TGCT, THCA, UCEC datasets (Firehose Legacy) from the cBioportal, https://www.cbioportal.org/ (accessed on 15 October 2023); and GDC PANCAN dataset from the USCS Xena Browser, https://xena.ucsc.edu/ (accessed on 22 October 2023); Prognoscan: all available datasets at http://dna00.bio.kyutech.ac.jp/PrognoScan/ (accessed on 8 May 2024).

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
