# Peer review of "The Epigenetic Modifiers HDAC2 and HDAC7 Inversely Associate with Cancer Stemness and Immunity in Solid Tumors"

_ijms, 2024, doi:10.3390/ijms25147841_

Round 1

Reviewer 1 Report

Comments and Suggestions for Authors

Dear authors:

The article with genomic studies illustrated that HDAC2 plays role in cancer cell stemless phenotype regulation, while HDAC7 perform more cancer cell regulation against stem cell phenotype. 

1. It is not clear what the data 'tumor types' stands for. Does it mean the HDAC2 and HDAC7 expression level in cells of the tumor microenvironment, or just the cancer cells? Please be specified.  HDAC7 seems to perform guardian role in T cell activity (https://doi.org/10.1101/2022.09.18.508452; https://doi.org/10.4049/jimmunol.1001179, etc.). If the genomic data is a mixture of cancer cell and T cells, this data does not stand well with the conclusion.

2. Overall, the statement that HDAC2 and HDAC7 is doing opposite roles is too simplified. The authors have over-stated the relationships. For example, we do not see many negative correlations between HDAC2 and HDAC7 in Figure 8 at the CD8+ T cell infiltration among different tumors. I do observe that HDAC7 plays negative role against HDAC2 in dendritic cell activation and CD4 infiltrations. However, it is still tumor to tumor. Instead of emphasizing the 'opposite' role, I would suggest using 'different'.

3. The conclusion that inhibiting HDAC2 and increase HDAC7 expression could benefit ICB is straightforward, but need more discussion. HDAC2 does contribute to ligand expression suppression and you can cite 'https://www.nature.com/articles/s41419-021-04047-2'. Many HDAC-I family inhibitors could contribute T cell infiltrations, and more citations could be easily found. Please cite as well. But, the role HDAC7 is unclear due to the lack of confirmation of my question #1.

4. the figure s12 showed a promising association between chemokine up regulation and HDAC7 expression. This looks solid and promising and should really be in the main manuscript that HDAC7 is contributing to immune response. 

5. Figure 9 the responders level of HDAC7 is higher than non-responder. It is not enough to prove that HDAC7 is associating with immune response. First, the patient data is too small. Second, the expression level does not align with activity level. more other data such as ATAC-seq should also be analyzed.

Overall, the author has demonstrated some novel findings through the genomic study. It is a scientific significant finding but requires more careful analysis. Look forward to the revision.

Author Response

We thank the Reviewer for his/her time and criticisms. It has certainly helped us improve the quality and focus of our manuscript. All the changes made to the manuscript are marked with the "track changes" function.

  1. It is not clear what the data 'tumor types' stands for. Does it mean the HDAC2 and HDAC7 expression level in cells of the tumor microenvironment, or just the cancer cells? Please be specified. HDAC7 seems to perform guardian role in T cell activity (https://doi.org/10.1101/2022.09.18.508452; https://doi.org/10.4049/jimmunol.1001179, etc.). If the genomic data is a mixture of cancer cell and T cells, this data does not stand well with the conclusion.

    Presented expression levels are mostly cancer cells as TCGA samples as a selection rule consist of at least 80% tumor nuclei. We were aware of this during manuscript preparation, and thus, we could derive presented HDAC7 conclusions with high confidence. It is known that other cell types (stroma, immune cells) may also be present in samples (although less than 20% per sample), but we see that both cancer cells and the tumor microenvironment express high-HDAC7. As presented below, the level of HDAC7 in cancer cell lines (blue: Broad: a collection of 917 cancer cell lines from the Cancer Cell Line Encyclopedia, GSE36133; Wappet: a collection of 627 cancer cell lines from AstraZeneca, GSE57083; Pommier: a collection of 174 cancer cell lines, from the NCI-60 panel; GSE32474) and in immune cells (orange: Lauwerys: a collection of CD4+ T cells and B cells, n = 49 samples, GSE4588; Matthes: a collection of 9 classes of leukocytes, n = 33 samples, GSE28491) is comparable (data analysis performed with R2: Genomics Analysis and Visualization Platform).

The level of HDAC7 in cancer cell lines

The following sentence has been added to the Methods section 4.1 to emphasize the information regarding the composition of tumor samples.:
Samples within individual studies (namely, tumor types) included in the TCGA come from primary, untreated tumor resection fragments which are composed of at least 80% tumor nuclei [80].

  1. Overall, the statement that HDAC2 and HDAC7 is doing opposite roles is too simplified. The authors have over-stated the relationships. For example, we do not see many negative correlations between HDAC2 and HDAC7 in Figure 8 at the CD8+ T cell infiltration among different tumors. I do observe that HDAC7 plays negative role against HDAC2 in dendritic cell activation and CD4 infiltrations. However, it is still tumor to tumor. Instead of emphasizing the 'opposite' role, I would suggest using 'different'.

The whole manuscript has been revised to soften this observation - the opposite has been changed to different/distinct etc. not to overstate the observed relationship. However, we still believe that our observation of HDAC7 tumor immune association is meaningful due to the strong correlation between HDAC7 and cytokine (including their receptors) expression. Due to the presence of cytokines, immune cells might be guided to enter malignant tissues. Thus, high-HDAC7 expression may contribute to immune cell infiltration into tumors. This pattern is not that clear when it comes to HDAC2, suggesting a non-immune-related role.

  1. The conclusion that inhibiting HDAC2 and increase HDAC7 expression could benefit ICB is straightforward, but need more discussion. HDAC2 does contribute to ligand expression suppression and you can cite 'https://www.nature.com/articles/s41419-021-04047-2'. Many HDAC-I family inhibitors could contribute T cell infiltrations, and more citations could be easily found. Please cite as well. But, the role HDAC7 is unclear due to the lack of confirmation of my question #1.

According to the Revierwer’s suggestions, we discussed the issue of HDAC inhibitors and ICI as a combined therapy in more details. In the current version of Discussion, the last paragraph ends as follow (additional text is marked in green).

Taking that into consideration and based on our results, we suggest that direct inhibition of class I HDAC family members (which are significantly overexpressed in stem cell-like, low-infiltrated solid tumors), together with immune checkpoint inhibitors, might result in a better outcome of treated patients, presumably by abolishing the self-renewal properties of cancer cells and by strengthening the anti-tumor immune responses. Our observations stay in line with previously reported enhanced antitumor immunity in triple-negative breast cancers achieved by HDAC2 knockout in the breast cancer mice models. As presented by Xu P. et al. [ref], HDAC2 is required for the chromatin remodeling of IFNγ-induced PD-L1 expression in breast tumors, and direct HDAC2 targeting suppresses immune escape of the tumor. Also, Zheng H. et al. [ref] have found that class I HDAC inhibitor romidepsin induced a strong T-cell-dependent antitumor response and enhanced the therapeutic effect of PD-1 inhibitors in lung adenocarcinoma. Later on, Orillion A. et al. [ref] reported the enhancement of the antitumor effect of PD-1 inhibition by entinostat - another class I HDAC inhibitor, in murine models of lung and renal cell carcinomas. Recently, Han R. et al. [ref] have elegantly summarized the rationale of targeting HDAC2 and immune checkpoint inhibitors in hepatocellular carcinomas. This novel tumor treatment strategy is endowed with great clinical application and research prospects, which provides a new opportunity to improve the overall prognosis of cancer patients further. Therefore, we suggest that HDAC2-high-expressing cancer patients who exhibit stem cell-like tumor phenotype might benefit from the combination therapy, including both class I HDACi and immune checkpoint inhibitors.”

  1. the figure s12 showed a promising association between chemokine up regulation and HDAC7 expression. This looks solid and promising and should really be in the main manuscript that HDAC7 is contributing to immune response.

Figure S12 has been split apart. In the main manuscript, we include Figure 9 (the former Figure 9 is now Figure 10) which consists of the former Figure S12BD (expression correlation of HDAC7 with chemokines and their receptors). Figure S12 now has two panels (A-B), the former Figure S12AC (expression correlation of HDAC2 with chemokines and their receptors). Thus, HDAC7 findings regarding this analysis are now in the main manuscript, while HDAC2 ones are left in the supplementary materials.

  1. Figure 9 the responders level of HDAC7 is higher than non-responder. It is not enough to prove that HDAC7 is associating with immune response. First, the patient data is too small. Second, the expression level does not align with activity level. more other data such as ATAC-seq should also be analyzed.

We agree that ATAC-seq could be a valuable contribution to support our findings. However, there is a lack of ready-to-use bioinformatics software which we could employ. Moreover, public ATAC-seq data availability specifically for HDAC7 is limited. Now at the end of the Introduction section, we emphasize our vision for the study's methodological approach and associated study limitations. We aimed to perform meaningful research based on already-existing and out-of-the-box data analysis solutions, alongside previously-published datasets, enforcing the idea that meaningful biological findings can be derived without the need to collect and analyze samples from scratch which is highly time-consuming and expensive. Unfortunately, due to the reasons above, we do not have enough funding to perform ATAC-seq analysis. However, according to your suggestion, the description of findings in the Results section 2.6. has been softened to emphasize more 1) limited patient cohorts, and 2) an indicative role of these findings, suggesting that further analysis in this direction should be performed, including the collection of additional samples. We believe that the inclusion of such promising possible results suggestions may direct other scientists in the experimental design of their future HDAC-related studies. Also, the following sentence has been added to the Results section 2.6.:
As patient cohorts in this analysis are too small, we underline that these findings are only suggestive, indicating the possible and promising direction of further research. Additional analyses should also be included to definitely prove the HDAC2/7 associations with immune response. In particular, HDAC2/7 activity and protein levels should be measured, alongside their chromatin accessibility.

Reviewer 2 Report

Comments and Suggestions for Authors

The manuscript from Maciejewski et al is an investigation on the possible correlation of members of the HDAC family with cancers stemness and immunity. I find this article intersting but according to my opinion some issues should be taken into consideration before publication.

1. The Introduction section is too long and should be smaller. Please try to reduce it keeping only the necessary information describing the HDAC family as well as the importance of investigating cancer stemness and antitumor immunity. The results and conclusion of the manuscript are also presented at the end of this section, although the do not belong there.

2. Please do not use abbrevations throughout the manuscript. An explanation in the suppl. file of some abbrevations is not sufficient and makes it difficult to follow the manuscript. It can be assumed for example that LUAD means lung adenocarcinoma, but the scientific merit of the article can be lost when the reader must made assumptions.

3. Please provide a list of the 22 solid tumors investigated in the printed form of the family and not as a supplementary file. It is important to know exactly what type of tumors here analysed. 

4. Results, Section 2.1. A differential expression of members of class II, III and IV family compared to normal tissue is observed. Please provide more information on this subject. A table could be helpful to illustrate your results.

5. Results, Section 2.2, line 172: the term most robustly is used to explain the correlation of members of the class I with staging but the direction of this correlation is not given.

6. Results, Section 2.2 and 2.5: Please provide Tables summyrising the results that you are describing. 

Author Response

We thank the Reviewer for his/her time and criticisms. It has certainly helped us improve the quality and focus of our manuscript. All the changes made to the manuscript are marked with the "track changes" function.

  1. The Introduction section is too long and should be smaller. Please try to reduce it keeping only the necessary information describing the HDAC family as well as the importance of investigating cancer stemness and antitumor immunity. The results and conclusion of the manuscript are also presented at the end of this section, although the do not belong there.

The Introduction section has been reduced according to the Reviewer’s suggestions. At the end of the section, we now include only a brief explanation of the study's vision and general methodological approach.

  1. Please do not use abbrevations throughout the manuscript. An explanation in the suppl. file of some abbrevations is not sufficient and makes it difficult to follow the manuscript. It can be assumed for example that LUAD means lung adenocarcinoma, but the scientific merit of the article can be lost when the reader must made assumptions.

    As for the Reviewer’s suggestion #3, we have now included the table with all study abbreviations in the main article, introduced in the first sentence of the Results section (so before any study abbreviation is mentioned). Moreover, the reader is invited to revise the list of employed abbreviations in the description of Figure 1 for the first time they encounter them. These changes should make the article with abbreviations easier to follow. In our opinion, removing the abbreviations would make all the sentences employing them unnecessarily long and would reduce the text's readability. Moreover, it has become a standard practice in TCGA-based studies to use suggested by the TCGA itself study/tumor abbreviations.
  2. Please provide a list of the 22 solid tumors investigated in the printed form of the family and not as a supplementary file. It is important to know exactly what type of tumors here analysed.

Table S1 has been moved into the main article as Table 2 now (in the place where Table 1 was originally placed). Table 1 has been moved from the Results into the Introduction section because 1) the breakdown of HDAC classes fits more there, and 2) two tables and a figure altogether introduced in a single sentence would be too many.

  1. Results, Section 2.1. A differential expression of members of class II, III and IV family compared to normal tissue is observed. Please provide more information on this subject. A table could be helpful to illustrate your results.

    The first sentence of this section has been updated to emphasize groups being compared in the differential expression analysis. This information is also included in the figure title and description.
    Previous: Firstly, we analyzed the expression of histone deacetylase (HDAC) gene family members, classified into five groups (Table 1), in selected solid tumors (Table S1) and relevant adjacent normal tissues using the TCGA [23] and GTEx data [24] (Figure 1), respectively.
    Now: Firstly, we analyzed the differential expression of histone deacetylase (HDAC) gene family members, classified into five groups (Table 1), in malignant tissues of selected solid tumors (Table 2), relative to adjacent normal tissues using the TCGA [23] and GTEx data [24], respectively (Figure 1).

    We have already specified details on the differentially expressed genes being observed, including their direction (up/down-regulation in malignant tissue compared to normal one). The reference of the mentioned table has been updated accordingly for your comment #3. As our results are originally presented by the heatmap (Figure 1) which indicates both correlation trend (continuous color scale) and statistical significance (symbols on each cell), we strongly believe that the presentation of exactly the same results in a table would be less attractive for the reader as 1) presenting the same results twice is an unnecessary redundancy, and 2) heatmap is an extended version of a table per se as it includes visual aid for results interpretation which omits the reader’s need to carefully analyze values in a table. Presenting only selected results in a table would infer the full picture of our results.

  2. Results, Section 2.2, line 172: the term most robustly is used to explain the correlation of members of the class I with staging but the direction of this correlation is not given.

The sentence in line 172 has been updated.

Previous: It is worth noticing that the expression of HDAC1, HDAC2, and HDAC3 (all members of class I) significantly correlates with TGCT’s staging most robustly among all correlations (Figure 3B).
Now: It is worth noticing that the expression of HDAC1, HDAC2, and HDAC3 (all members of class I) significantly correlates with TGCT’s staging most robustly among all correlations in a positive manner (Figure 3B).

  1. Results, Section 2.2 and 2.5: Please provide Tables summyrising the results that you are describing.

Section 2.2 introduces figures 3 (heatmap) and S4 (bubble plot). As for your comment #4, we believe that the presentation of exactly the same results in a table would be less attractive for the reader for described before reasons, which are: lower readability, results redundancy, and in the case of including in a table only partial results - lower transparency and possible results misinterpretation by the reader.

Section 2.5 introduces figures 8, and 9 (added figure to comment #4 of Reviewer 1) and S10-S15. As these figures are mostly heatmaps, we maintain our statement on not presenting heatmap information as tables, as reasoned in our response to your comments #4 and #6 above. Moreover, as this section introduces many various results with many exceptions to general observed trends, any generalization in a table form would create a risk of overstating and thus results misinterpretation by overrepresentation of actual findings, beyond the limitations of our study.

Round 2

Reviewer 2 Report

Comments and Suggestions for Authors

Thank you for addressing all my comments.